# Special Vehicle Detection from UAV Perspetive via YOLO-GNS Based Deep Learning Network

**Zifeng Qiu** [1,2,3], **Huihui Bai** [1] **and Taoyi Chen** [3,*]

1 Institute of Information Science, Beijing Jiaotong University, Beijing 100044, China
2 Key Laboratory of Aerospace Information Applications of CETC, Shijiazhuang 050081, China
3 The 54th Research Institute of CETC, Shijiazhuang 050081, China
* Correspondence: 20112012@bjtu.edu.cn

**Abstract:** At this moment, many special vehicles are engaged in illegal activities such as illegal mining, oil and gas theft, the destruction of green spaces, and illegal construction, which have serious negative impacts on the environment and the economy. The illegal activities of these special vehicles are becoming more and more rampant because of the limited number of inspectors and the high cost required for surveillance. The development of drone remote sensing is playing an important role in allowing efficient and intelligent monitoring of special vehicles. Due to limited onboard computing resources, special vehicle object detection still faces challenges in practical applications. In order to achieve the balance between detection accuracy and computational cost, we propose a novel algorithm named YOLO-GNS for special vehicle detection from the UAV perspective. Firstly, the Single Stage Headless (SSH) context structure is introduced to improve the feature extraction and facilitate the detection of small or obscured objects. Meanwhile, the computational cost of the algorithm is reduced in view of GhostNet by replacing the complex convolution with a linear transform by simple operation. To illustrate the performance of the algorithm, thousands of images are dedicated to sculpting in a variety of scenes and weather, each with a UAV view of special vehicles. Quantitative and comparative experiments have also been performed. Compared to other derivatives, the algorithm shows a 4.4% increase in average detection accuracy and a 1.6 increase in detection frame rate. These improvements are considered to be useful for UAV applications, especially for special vehicle detection in a variety of scenarios.

**Keywords:** drone; special vehicle; object detection; YOLO; SSH; GhostNet

## 1. Introduction

Special vehicles refer to motorized machines that are distinct from conventional automobiles in terms of their physical characteristics, such as shape, size, and weight. Those vehicles are typically used for a variety of purposes, including traction, obstacle removal, cleaning, lifting, loading and unloading, mixing, excavation, bulldozing, and road rolling, etc.

The detection of special vehicles in oil and gas pipelines [1], transmission lines [2], urban illegal construction [3], theft, and excavation scenarios is of great importance in order to ensure the security of these areas. This is because in the above scenarios, the presence of special vehicles often represents a high risk that these scenarios will occur, and the nature of special vehicles may cause damage to important property. The use of unmanned aerial vehicles to patrol and search for special vehicles in these scenarios has gradually become a mainstream application trend [4]. However, due to the particular shape of special vehicles, manual interpretation has low efficiency, high misjudgment, and omission. The application of a deep neural network in the automatic detection of special vehicles has been applied to some extent, but it is not mature yet, and the accuracy of existing methods is relatively poor.

Experts and scholars have proposed a variety of depth neural network methods for target detection in UAV aerial images including various vehicles. Various techniques

including CNNs, RNNs, autoencoders, and GANs have been used in vehicle detection and have yielded interesting results for many tasks [5]. To detect small objects, some techniques divide the last layer of the neural network into multiple variable-sized chunks to extract features at different scales, while other approaches remove the deeper layers of the CNN, allowing the number of feature points of the target to increase [6]. Liu W et al. proposed the YOLOV5-Tassel network, which combines CSPDarknet53 and BiFPN to efficiently extract minute features and introduces the SimAM attention mechanism in the neck module to extract the features of interest before each detection head [7]. Zhou H et al. designed a data augmentation method including background replacement and noise increase in order to solve the detection of tiny targets such as cars and planes, and constructed the ADCSPDarkent53 backbone network based on YOLO, which was used to modify the loss of localization function and improve the detection accuracy [8]. In order to solve the problems of low contrast, dense distribution, and weak features of small targets, Wang J et al. constructed corresponding feature mapping relations, solved the level of adjacency between misaligned features, adjusted and fused shallow spatial features and deep semantic features, and finally improved the recognition ability of small objects [9]. Li Q et al. proposed a "rotatable region-based residual network (R3-Net)" to distinguish vehicles with different directions from aerial images and used VGG16 or ResNet101 as the backbone of R3-Net [10]. Li et al. presented an algorithm for detecting sea targets based on UAV. This algorithm optimizes feature fusion calculation and enhances feature extraction at the same time, but the computational load is too large [11]. Wang et al. used the Initial Horizontal Connection Network to enhance the Feature Pyramid Network. In addition, the use of the Semantic Attention Network to provide semantic features helps to distinguish interesting objects from cluttered backgrounds, but how the algorithm performs as expected in complex and variable aerial images needs further study [12]. Mantau et al. used visible light and thermal infrared data taken from drones to find poachers. They used YOLOv5 as their basic network and optimized it using migration learning, but this method did not work well with the fusion of different data sources [13]. Deng et al. proposed a network for detecting small objects in aerial images. They designed a Vehicle Proposal Network, which proposed areas similar to vehicles [14]. Tian et al. proposed a bineural network review method, which classifies the secondary characteristics of the suspicious target area in the unmanned aerial vehicle image, quickly filters the missing targets in one-stage detection, and achieves high-quality detection of small targets [15].

In terms of drone inspection of vehicles, Jianghuan Xie et al. proposed an anchor-free detector, called residual feature enhanced pyramid network (RFEPNet), for vehicle detection from the UAV perspective. RFEPNet contains a cross-layer context fusion network (CLCFNet) and a residual feature enhancement module (RFEM) based on pyramid convolution to achieve small target vehicle detection [16]. Wan Y et al. proposed an adaptive region selection detection framework for the retrieval of targets, such as vehicles in the field of search and rescue, adding a new detection head to achieve better detection of small targets [17]. Liu Mingjie et al. developed a detection method for small-sized vehicles in drone view, specifically optimized by connecting two ResNet units with the same width and height and adding convolutional operations in the early layers to enrich the spatial information [18]. Zhongyu Zhang et al. proposed a YOLOv3-based Deeply Separable attention-guided network (DAGN) that combines feature cascading and attention blocks and improves the loss function and candidate merging algorithm of YOLOv3. With these strategies, the performance of vehicle detection is improved while sacrificing some detection speed [19]. Wang Zhang et al. proposed a novel multiscale and occlusion-aware network (MSOA-Net) for UAV-based vehicle segmentation, which consists of two parts, including a multiscale feature adaptive fusion network (MSFAF-Net) and a region-attention-based three-headed network (RATH-Net) [20]. Xin Luo et al. developed a fast automatic vehicle detection method for UAV images, constructed a vehicle dataset for target recognition, and proposed a YOLOv3 vehicle detection framework for relatively small and dense vehicle targets [21]. Navaneeth Balamuralidhar proposed MultEYE that can detect, track, and

estimate the velocity of a vehicle in a sequence of aerial images using a multi-task learning approach with a segmentation head added to the backbone of the object detector to form the MultEYE object detection architecture [22].

When drones patrol oil and gas pipelines, power transmission lines, urban violations and other fields, the size of special vehicles in the images change greatly, and there are many small targets. The feature information carried by camera overhead is limited and changeable, which increases the difficulty of detection. Secondly, the UAV cruises across complex and changeable scenes such as cities, wilderness, green areas, bare soil, and so on. Some areas contain dense targets, which makes it difficult to distinguish some similar objects. Finally, the shooting angle also brings more noise interference, and the special vehicle will be weakened, obscured, or even camouflaged, unable to expose the characteristics of the target. Due to the characteristics of variable target scale, a number of small targets, and the complex background of special vehicles, it is difficult to meet the requirements of speed and accuracy for patrol tasks if the above research methods are directly applied to special vehicle detection from a UAV perspective.

In order to solve the problem of special vehicle detection in complex backgrounds from the perspective of drones, we propose a deep neural network algorithm (YOLO-GNS) based on YOLO and optimized by GhostNet (GN) and Single Stage Headless (SSH), which can be used to detect special vehicles effectively. Firstly, the SSH network structure is added behind the FPN network to parallel several convolution layers, which enhances the convolution layer perception field and extracts the high semantic features of the special vehicle targets. Secondly, in order to improve the detection speed to meet the requirements of UAV, the GPU version of GN (G-GN) is used to reduce the computational consumption of the network. Finally, we have searched for a large number of rare places to take aerial photos and created a dataset containing a large number of special vehicle targets. We have experimented with YOLO-GS on the special vehicle (SEVE) dataset and public dataset to verify the effectiveness of the proposed method.

The rest of this paper is arranged as follows. Section 2 describes the proposed target detection method YOLO-GNS and the necessary theoretical information. Section 3 introduces the special data sets, evaluation methods, and detailed experimental results. In Section 4, we draw conclusions and determine the direction of future research.

## 2. Materials and Methods

### 2.1. Principle of YOLOv7 Network Structure

YOLO (You Only Look Once) is a one-stage target detection algorithm based on regression method proposed by Redmon et al. It has been developed into several versions [23–29]. As the latest upgrade of YOLO series, YOLOv7 has been improved from data enhancement, backbone network, activation function, and loss function, so that it has higher detection accuracy and faster detection speed.

The YOLOv7 algorithm employs strategies such as extended efficient long-range attention network (E-ELAN), Concatenation-Based models, and convolution parameterization to achieve a good balance between detection efficiency and accuracy.

As shown in Figure 1, YOLOv7 network is composed of four parts: Input, Backbone, Neck, and Head.

The Input section scales the input image to a uniform pixel size to meet the input size requirements of the backbone network. The Backbone part is composed of several CBS modules, E-ELAN modules, and MP1 modules. The CBS module is composed of convolution layer, batch normalization layer (BN), sigmoid-weighted linear unit activation function to extract image features at different scales, as shown in Figure 2.

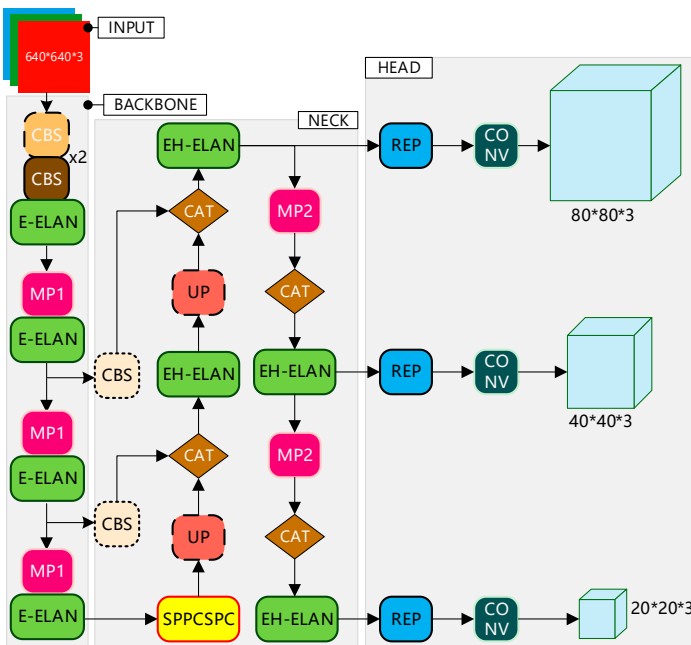

**Figure 1.** The original structure of yolov7.

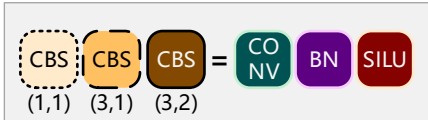

**Figure 2.** The structure of CBS module.

ELAN module consists of several CBS modules, whose input and output feature sizes remain the same. By guiding the computing blocks of different feature groups to learn more diverse features, the learning ability of the network is improved without destroying the original gradient path, as shown in Figure 3.

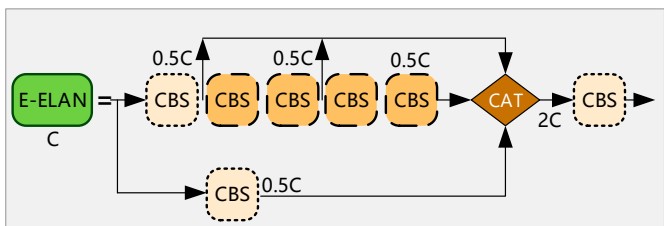

**Figure 3.** The structure of E-ELAN module.

MP1 module adds Maxpool layer on the basis of CBS module, which constitutes the upper and lower branches. The upper branch halves the image length and width through Maxpool and the image channel through CBS module. The lower branch halves the image channel through the first CBS module; the second CBS layer halves the image length and width and finally uses the Cat operation to fuse the features extracted from the upper and lower branches, which improves the feature extraction ability of the network, as shown in Figure 4.

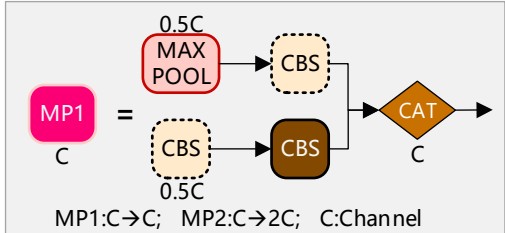

**Figure 4.** The structure of MP1module.

The Neck part is composed of Path Aggregation Feature Pyramid Network (PAFPN) structure, mainly including SPPCSPC module, ELAN-H module, and UP module. By introducing the bottom-up path, the bottom-level information can be easily transferred to the top level, which enables the efficient fusion of different hierarchical features.

The SPPCSPC module is mainly composed of CBS module, CAT module, and Maxpool module, which get different perception fields through maximum pooling, as shown in Figure 5.

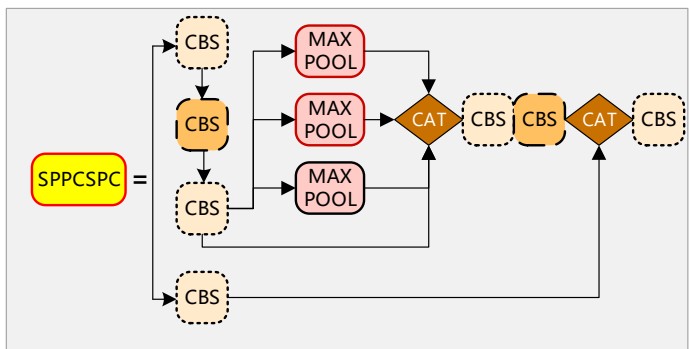

**Figure 5.** The structure of SPPCSPC module.

EH-ELAN module is similar to E-ELAN module but slightly different in that it selects five branches to add up with different number of outputs, as shown in Figure 6.

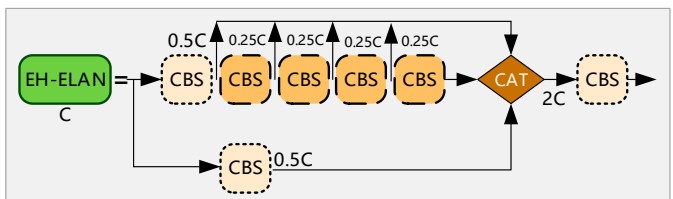

**Figure 6.** The structure of EH-ELAN module.

The UP module is composed of CBS and up sampling modules, as shown in Figure 7.

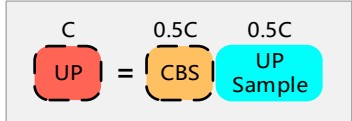

**Figure 7.** The structure of UP module.

Head adjusts the number of image channels for three different scales of Neck output through RepVGG Block (REP) structure, and then passes through $1 \times 1$ Convolution is used for predicting confidence, category, and anchor frame.

The REP structure is divided into train and deploy versions, as shown in Figure 8. The train version has three branches. The top branch is $3 \times 3$ convolution, which is used for

feature extraction; the middle branch is $1 \times 1$ convolution, which is used for smoothing features; and the bottom branch is an Identity, which is moved without convolution and finally added together. The deploy version contains a $3 \times 3$ convolution with a stride of 1, which is converted from the training module parameterization.

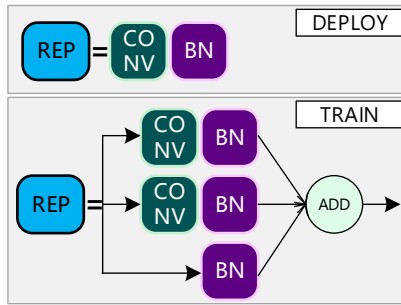

**Figure 8.** The structure of REP module.

Although the YOLOv7 algorithm framework performs well in common task scenarios, such as pedestrian and normal vehicle detection, there are still many problems when it is applied directly to the detection of special vehicles from the perspective of unmanned aerial vehicles: (1) Compared with common scenarios, the target scale in unmanned aerial vehicle images changes more, and there are more small targets, which further increases the difficulty of special vehicle detection; (2) The background of the scene in which the special vehicle is located is complex, and there is no corresponding context mechanism to handle the complex background, which results in the inaccurate detection of the special vehicle in the complex background; (3) UAV images require higher detection speed, but conventional YOLOv7 does not have the detection acceleration function for UAV. To solve the above problems, the algorithm in this paper is based on YOLOv7 and improved.

### 2.2. YOLO-GNS Algorithm

This section introduces the special vehicle target detection algorithm from the perspective of UAV, as shown in Figure 9. With YOLOv7 as the framework, the Backbone is improved based on GhostNet to enhance the feature extraction ability and improve the detection speed; in the view of UAV, it is beneficial to detect the weakened or occluded special vehicles from the complex scene. In order to improve the ability to detect small targets, SSH modules are added behind the pafpn structure of yolov7 to merge context information. Therefore, the algorithm is named YOLO-GNS. Compared with YOLOv7 and other derivatives, YOLO-GNS can achieve the best balance between detection accuracy and calculation cost.

2.2.1. Improvement of Backbone Network Based on GhostNet

In the backbone network of the original YOLOv7, due to the high redundancy of the intermediate feature map calculated by a large number of conventional convolutional CBS modules, the computing cost will increase. YOLO-GNS built GhostMP and GhostELAN modules to form a backbone network to extract UAV image features by drawing on the ideas of GhostNet [30]. GhostNet has the advantages of maintaining the recognition performance of similarity and reducing the convolution operation at the same time, which can greatly reduce the number of model parameters while maintaining high performance.

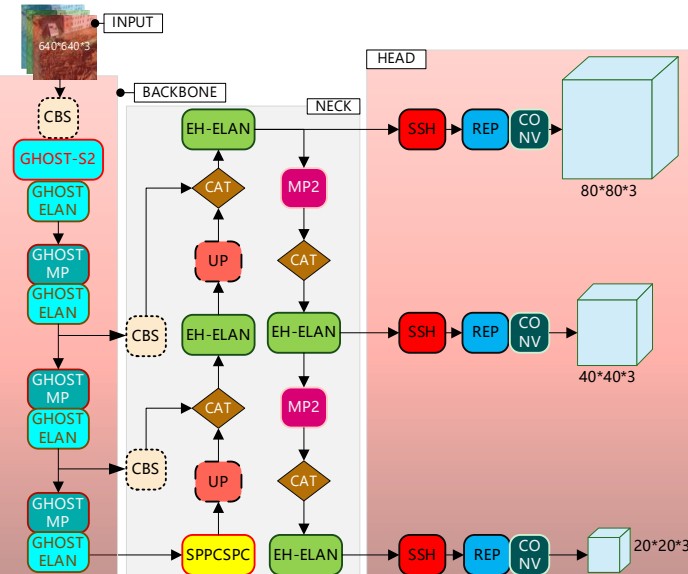

**Figure 9.** The structure of YOLO-GNS algorithm.

The GhostMP module is composed of Maxpool, GhostS2, CBS, CAT. The GhostELAN module is composed of GhostS1, GhostS2, CBS, and CAT. GhostS1 consists of two stacked Ghost convolutions (Ghost Conv), the first Ghost Conv increasing the number of channels and the second Ghost Conv reduces the number of channels to match the shortcut path, making the number of channels for the input signature map the same as which in the output signature map for the second Ghost Conv. The shortcut path of GhostS2 is implemented by depth-wise convolution (DW Conv) with a downsampling layer and a stride = 2 to reduce the number of channels. Add represents a signature graph addition operation where the number of channels does not change.

The implementation of GhostConv is divided into three steps: the first step is to use ordinary convolution calculation to get a feature map with less channel information, the second step is to use inexpensive operation to generate more feature maps, and the last step is to connect different feature maps to form a new output.

In ordinary convolution, given input data $X \in R^{c \times h \times w}$, $c$ denotes the number of input channels; $h$ and $w$ denote the height and width of the input data, respectively, and are used to generate any convolution layer of N feature map, as shown in Equation (1):

$$Y \in X^* f + B \tag{1}$$

where: $*$ is a convolution operator, $B$ is a deviation term, $Y \in R^{h' \times w' \times n}$ represents the output feature map of N channels, $f \in R^{c \times k \times k \times n}$ is the convolution kernel size in a convolution layer, $h'$ and $w'$ represent the height and width of the output data, respectively, $k \times k$ denotes the size of the convolution kernel $f$. In ordinary convolution operations, because the number of convolution cores $n$ and channel $c$ is very large, the number of FLOPs required is $n \cdot h' \cdot w' \cdot c \cdot k \cdot k$.

Thus, the parameters to be optimized for operation ($f$ and $B$) are determined by the size of the input and output feature maps. Since the output feature maps of ordinary convolution layers are usually redundant and may have similar redundancy to each other, it is not necessary to use a large number of parameters FLOP to generate redundant feature maps, which are "Ghost" converted from a few original feature maps by some inexpensive linear operations. These original feature maps are usually generated by ordinary convolution kernels and have less channel information. Generally, $m$ original feature map $Y' \in R^{h' \times w' \times m}$ is generated by once convolution:

$$Y' = X^* f' \tag{2}$$

where: $f' \in R^{c \times k \times k \times m}$ is a convolution kernel, $m \leqslant n$. To maintain the same spatial size as the output feature map, the hyperparametric (convolution size, stride, padding) is the same as the ordinary convolution. To further obtain the required $n$ feature maps, a series of inexpensive linear operations are used for each original feature in $Y'$, resulting in $s$ Ghost feature maps, as shown in Formula (3):

$$y_{ij} = \phi_{i,j}\left(y_i'\right); \forall i = 1, 2, \cdots, m, j = 1, 2, \cdots, s \tag{3}$$

where:$y_{ij}$ represents the first primitive feature map in $Y'$. $\phi_{i,j}$ represents the $j$th linear operation used to generate the $j$th Ghost feature graph. By using inexpensive linear operations, we can get $n = m \cdot s$ feature maps as output of the Ghost module, as shown in Formula (4):

$$Y = [y_{11}, y_{12}, \cdots, y_{ms}] \tag{4}$$

The Ghost module divides the original convolution layer into two phases, as shown in Figure 10. The first phase uses a small number of convolution cores to generate the original feature map, and the second phase uses inexpensive transformation to generate more Ghost feature maps. Linear operations are used on each channel to reduce computational effort.

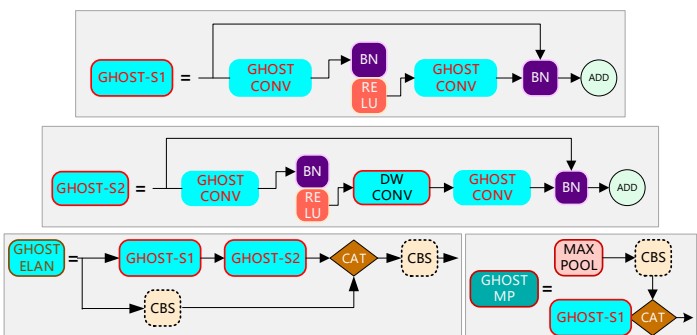

**Figure 10.** The structure of GhostNet in YOLO-GNS.

### 2.2.2. Prediction Optimization Based on SSH Structure

In order to improve the small target detection ability and further shorten the inference time, Single Stage Headless (SSH) algorithm [31] is introduced into the network, which is a single-stage context network structure. The two-stage context network structure combines more context information by increasing the size of the candidate box. Nevertheless, SSH combines context information through a single convolution layer, where the Context-Network structure of the SSH detection module is shown in Figure 11, which requires less memory to detect and locate more accurately.

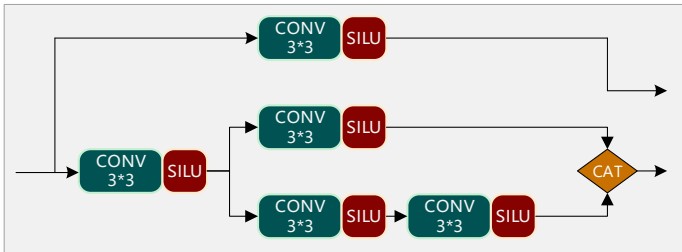

**Figure 11.** The structure of Context-Network in SSH.

In YOLO-GNS, add the SSH context network structure before the REP structure. First, reduce the number of channels to X/2 through $3 \times 3$ convolution layer and SILU activation function (3C-SILU), and then send this result to two branches. One branch contains only one 3C-SILU operation, which results in the feature that the channel is X/2. The other

branch contains two consecutive 3C-SILU operations, which also results in the feature that the channel is X/2. Finally, concatenate the two feature maps to get the final output of the SSH context network structure.

The SSH context network structure incorporates more context information and is approximated by increasing the sensory field of the feature maps. For example, a small field can only see the special vehicle itself, while a larger field can see the excavator head, caterpillar, and other places.

Generally, deeper feature layers contain more abstract semantic information to facilitate classification, while shallow features have more specific information, such as edges, angles, and so on, to facilitate the positioning of bounding box.

Therefore, the SSH context network structure integrates the current and high-level feature information, effectively improves the detection ability of the weakened and obstructed special vehicles in complex environments, helps to improve the accuracy of the algorithm, and does not significantly increase the additional computational load.

## 3. Results

In order to evaluate the special vehicle detection performance of YOLO-GNS algorithm in this paper, this experiment conducts training and testing on special vehicle (SEVE) dataset. Additionally, to evaluate the general performance of the algorithm, this experiment adds training and testing on the Microsoft COCO dataset.

### 3.1. Special Vehicle Dataset

Heretofore, there is no public data set of special vehicles from the perspective of drones. Therefore, from January 2021 to June 2022, we used UAV to shoot a large number of videos at multiple heights and angles over construction areas, wilderness, building sites, and other areas. After that, frames are extracted and labeled from these videos to form a special vehicle dataset. This dataset contains 17,992 pairs of images and labels, including 14,392 training sets, 1800 validation sets, and 1800 test sets. The image resolution in SEVE dataset is 1920 × 1080. The types of special vehicles include cranes, traction vehicles, tank trucks, obstacle removal vehicles, cleaning vehicles, lifting vehicles, loading and unloading vehicles, mixing vehicles, excavators, bulldozers, and road rollers. The different scene types include urban, rural, arable, woodland, grassland, construction land, roads, etc. Some examples of the dataset are shown in Figure 12.

### 3.2. Experimental Environment and Settings

The experiment is based on 64-bit operating system Windows 10, the CPU is Intel Xeon Gold 6246R, the GPU uses NVIDIA GeForce RTX3090, and the deep learning framework is Pytorch v1.7.0. We use Frames Per Second (FPS) to measure the detection speed, which indicates the number of images processed by the specified hardware per second by the detection model. In the experiment, the FPS for each method is tested on a single GPU device. IOU is set to 0.5, The mAP (mean Average Precision), an index related to the IOU threshold, was used as the standard of detection accuracy. In multi-category target detection, the curve drawn by each category based on its accuracy (Precision) and recall (Recall) is called a P-R curve, in which the average recognition accuracy of a category is equal. AP@0.5 (Average Precision, IoU threshold greater than 0.5) is the size of the area below the P-R curve of this category. mAP@0.5 Average recognition accuracy by all categories AP@0.5 add up to get the average.

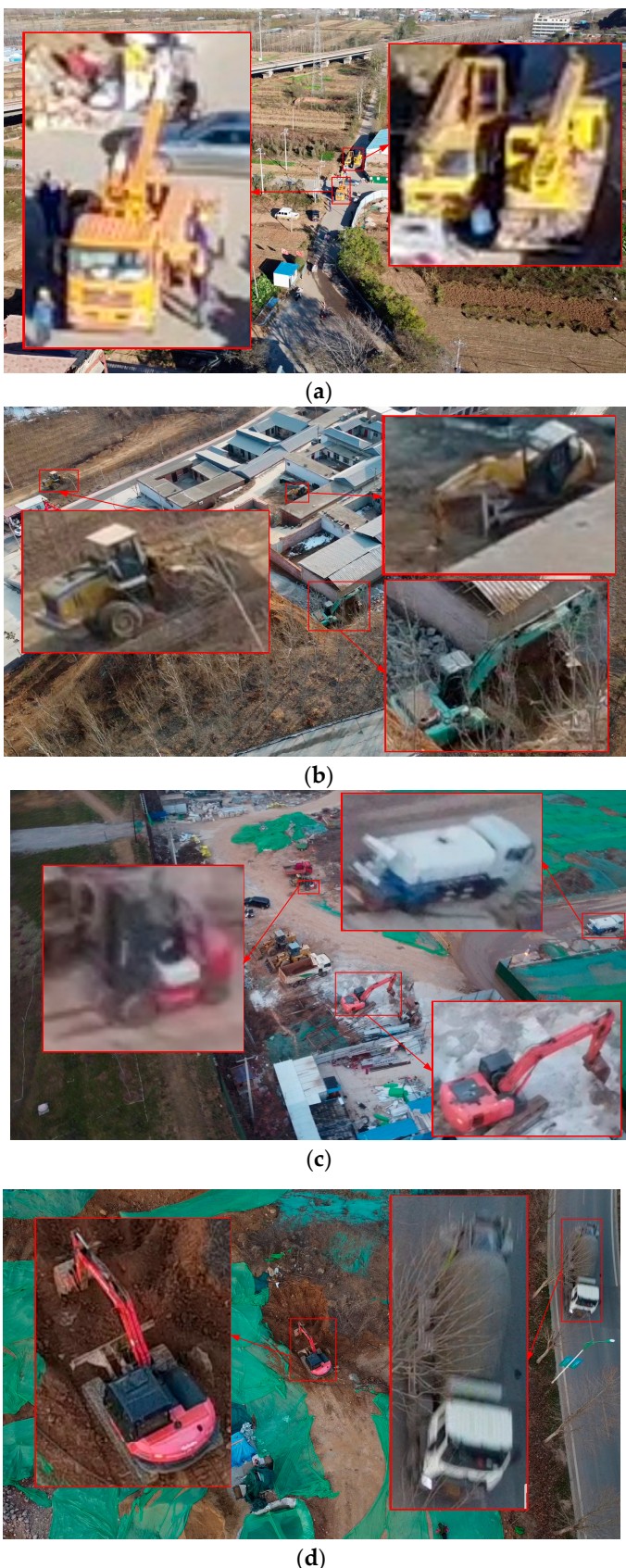

**Figure 12.** Sample Images of SEVE dataset. (**a**) Cranes in construction areas; (**b**) Excavator and loaders in building sites; (**c**) Forklifts in construction areas; (**d**) Excavator and tank trucks in wilderness.

Precision and recall are defined as:

$$Recall = \frac{TP}{TP + FN} \tag{5}$$

$$Precision = \frac{TP}{TP + FP} \tag{6}$$

$$AP = \int_0^1 P(R)dR \tag{7}$$

$$mAP = \frac{1}{c}\sum_{i=1}^{c} AP_i \tag{8}$$

Among them, *TP* was the real case, *FP* was the false positive case, *FN* was the false negative case, and *C* was the total number of categories detected for the target.

Due to the limitation of the experimental device, the input image size is scaled to $800 \times 800$ pixels. The optimizer uses SGD; the learning rate is $1 \times 10^{-2}$; the momentum is 0.9; the weight decay is $5 \times 10^{-4}$, using the Cosine Annealing algorithm to adjust the learning rate; the batch size is 8; and the training durations are 300 epochs, 10 training epochs, and 1 test epochs alternately.

### 3.3. Experimental Results and Analysis

This paper conducts experiments on the open dataset COCO and the SEVE dataset created in this paper to verify the validity of the proposed methods. The experiment is divided into three parts:

(1) Experiments are carried out on the SEVE dataset to verify the feasibility of the proposed method, and to compare the results with those of other target detection methods on this dataset to illustrate the advantages of this method;

(2) Verify the universality of this method on COCO datasets;

(3) Designing an ablation experiment further demonstrates the validity of the method.

### 3.3.1. Experiments on SEVE Dataset

In this experiment, the YOLO-GNS algorithm is compared with the prevailing target detection algorithms in the SEVE dataset created in this paper. The experimental results are shown in Table 1. Table 1 contains nine categories: C, L, T, M, F, P, R, EL, and EX, corresponding to the SEVE dataset and referring to cranes, loader cars, tank cars, mixer cars, forklifts, piling machines, road rollers, elevate cars, and excavators. The resulting data AP@0.5 represent the average recognition accuracy of this category under different methods, while data in column mAP@0.5 represents the average recognition accuracy of all categories. Params represent the size of the paraments of each method. The resulting data represent the average recognition accuracy for all categories for different datasets under different methods.

**Table 1.** Comparison of Detection Accuracy of Different Target Detection Algorithms on SEVE dataset.

| Methods | AP@0.5(%) | | | | | | | | | mAP@0.5 (%) | Params(M) | FPS |
|---|---|---|---|---|---|---|---|---|---|---|---|---|
| | C | L | T | M | F | P | R | EL | EX | | | |
| Faster-RCNN | 73.2 | 75.5 | 76.1 | 80.2 | 78.1 | 81.3 | 56.3 | 45.5 | 21.3 | 65.3 | 186.3 | 16.8 |
| RetinaNet | 77.5 | 78.6 | 85.1 | 82.3 | 81.5 | 80.6 | 57.6 | 49.1 | 23.5 | 68.4 | 28.5 | 19.5 |
| YOLOV4 | 78.7 | 80.1 | 82.3 | 83.5 | 82.6 | 78.3 | 60.5 | 55.8 | 30.3 | 70.2 | 64.4 | 25.6 |
| YOLOV5-X | 79.8 | 78.1 | 85.6 | 83.9 | 83.1 | 82.5 | 59.1 | 58.3 | 32.5 | 71.4 | 86.7 | 29.2 |
| YOLOV7 | 80.5 | 82.3 | 86.4 | 88.6 | 85.3 | 86.4 | 65.3 | 60.8 | 45.8 | 75.7 | 36.9 | 31.5 |
| YOLO-GNS | 85.9 | 86.9 | 89.4 | 91.3 | 90.1 | 89.6 | 69.5 | 67.3 | 50.8 | 80.1 | 30.7 | 33.1 |

In the SEVE dataset, special vehicle targets vary greatly in scale and there are mostly small targets. The image background is complex and volatile, and it is difficult to distinguish the targets into the background, and some targets are also obscured, which brings some difficulty to the detection. The improved network in this paper has significant accuracy advantages compared with other mainstream target detection algorithms. The method in this paper achieves the best results on the SEVE dataset with 80.1%, which is 4.4% higher accuracy compared to YOLOV7; meanwhile, the mAP is 14.8%, 11.7%, 9.9%, and 8.7% higher compared to four target detection algorithms, namely Faster R-CNN, RetinaNet, YOLOV4, and YOLOV5, respectively; although the YOLOv7 and YOLOv5 detection speeds are close to that of YOLO-GNS, the mAPs are all lower than the methods in this paper. Owing to GhostNet applied in the backbone section, the parameters of YOLO-GNS are reduced by 6.2M. In the case of low differentiation of YOLO series backbone networks, the mAP of this paper's method is higher and the detection speed is faster, which indicates that this paper's method makes up for the difference of backbone networks and reflects greater advantages. Due to the reconstructed backbone network and the parallel SSH context network that makes the network structure of this paper in the case of increasing complexity, the detection speed is not reduced and can meet the needs of engineering applications.

The detection results of YOLOV7 and this paper's method YOLO-GNS are shown in Figures 13–15. Column (a) shows the recognition results of the YOLO-GNS network, and column (b) shows the recognition results of the original YOLOV7 network. A comparison of the results of the two networks shows that the YOLO-GNS network in this paper has improved accuracy in terms of bounding box and category probabilities. On the other hand, the recognition of special vehicles, such as cranes, loader cars, tank cars, mixer cars, forklifts, and excavators, and their differences from ordinary vehicles are improved in the proposed model.

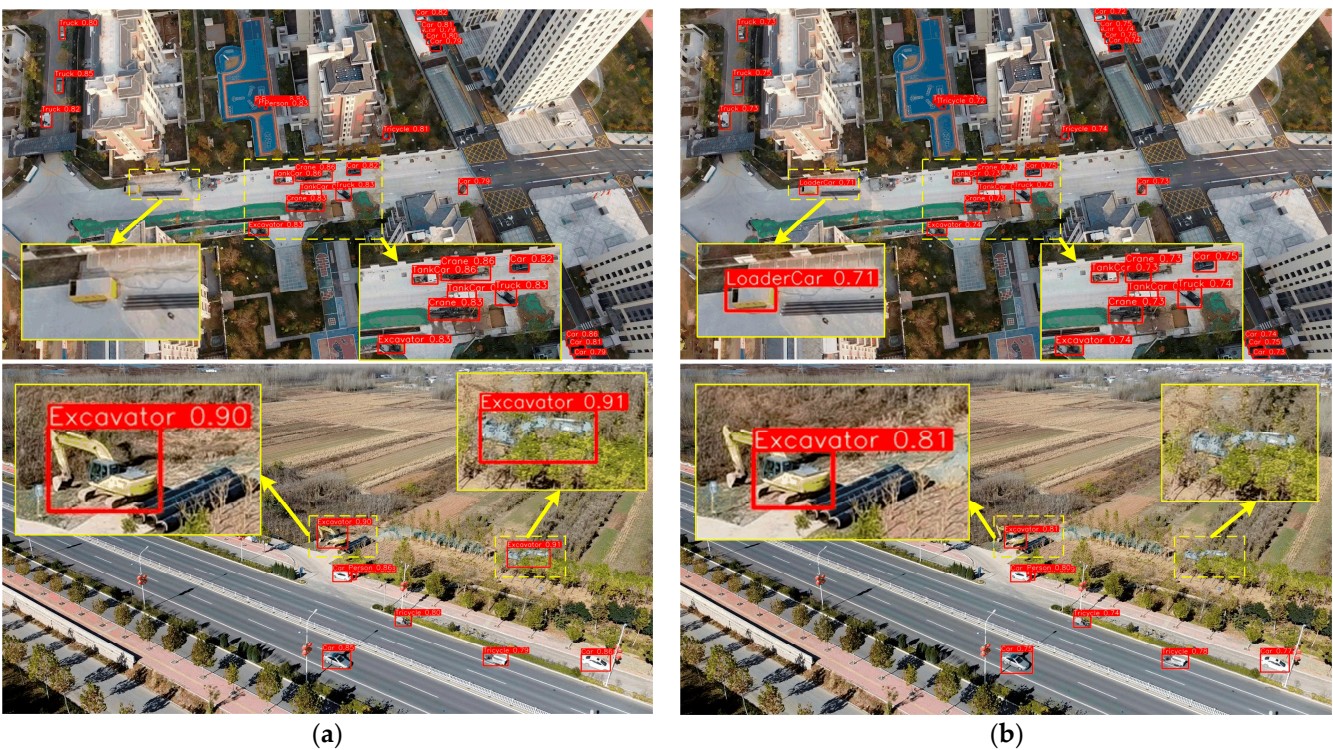

(**a**) (**b**)

**Figure 13.** Recognition results in crowded environments of SEVE Dataset. (**a**) Recognition results of the YOLO-GNS network; (**b**) Recognition results of the YOLO-V7 network.

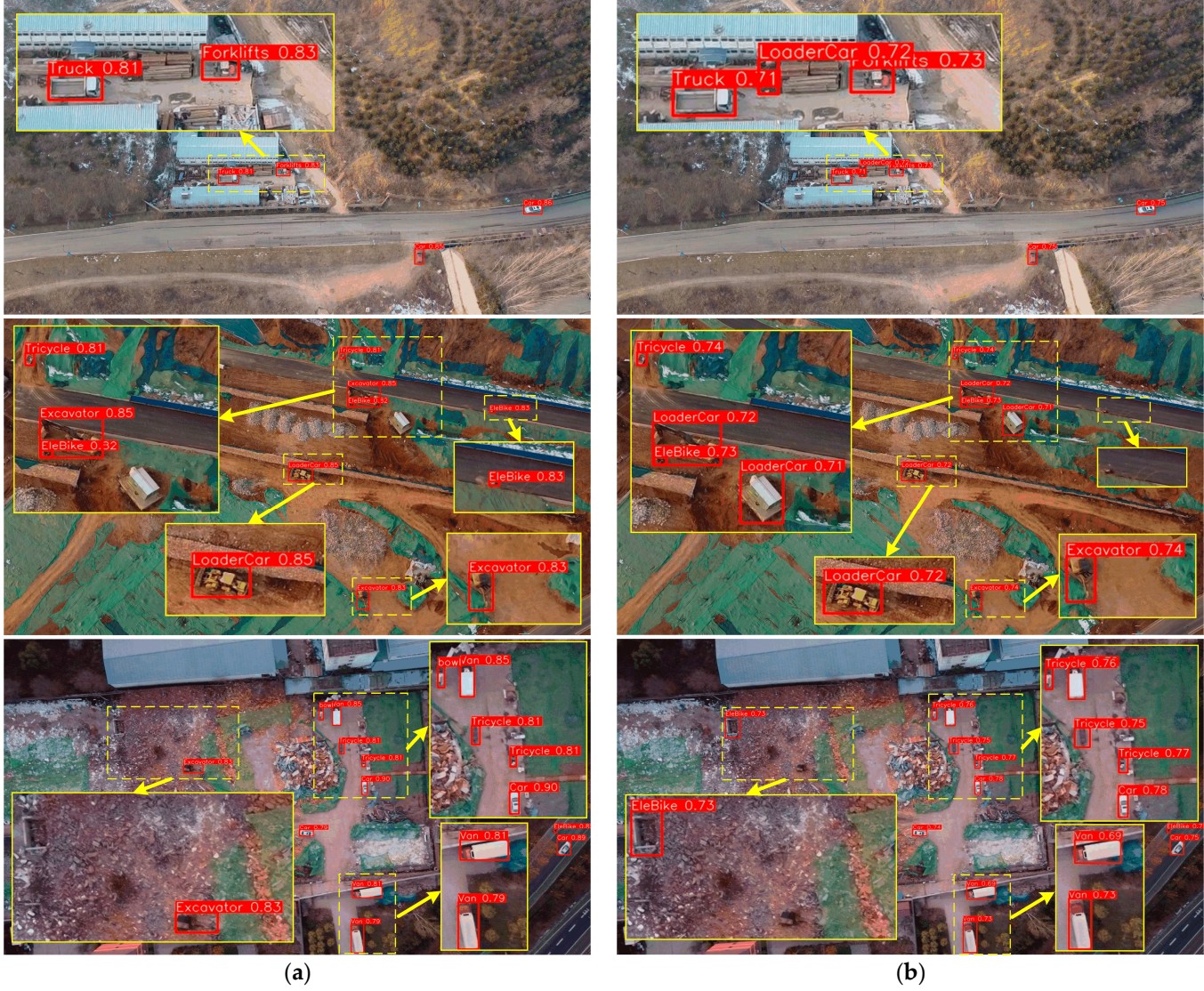

**Figure 14.** Recognition results in complex background of SEVE Dataset. (**a**) Recognition results of the YOLO-GNS network; (**b**) Recognition results of the YOLO-V7 network.

In Figure 13, it is shown that in crowded environments such as cities and roads, YOLO-GNS can identify obscured special vehicles and does not cause false detections, while YOLOV7 produces false detections and missed detections and has lower class probability values than the modified model. In Figure 14, it is shown that YOLO-GNS distinguishes special vehicles from ordinary vehicles by extracting smaller and more accurate features in environments with camouflage characteristics, such as construction sites, and can identify special vehicles that are highly similar to the background. In Figure 15, it is shown that the YOLO-GNS network is able to identify different special vehicle types in complex and challenging conditions under poor lighting conditions and bad weather, while the original YOLOV7 model would show quite a few missed and false detections. In conclusion, the YOLO-GNS proposed in this paper is able to identify targets with a high prediction probability under a variety of complex scenarios. In some cases, the base model YOLOV7 cannot accurately identify special vehicles, or it has a lower probability than YOLO-GNS.

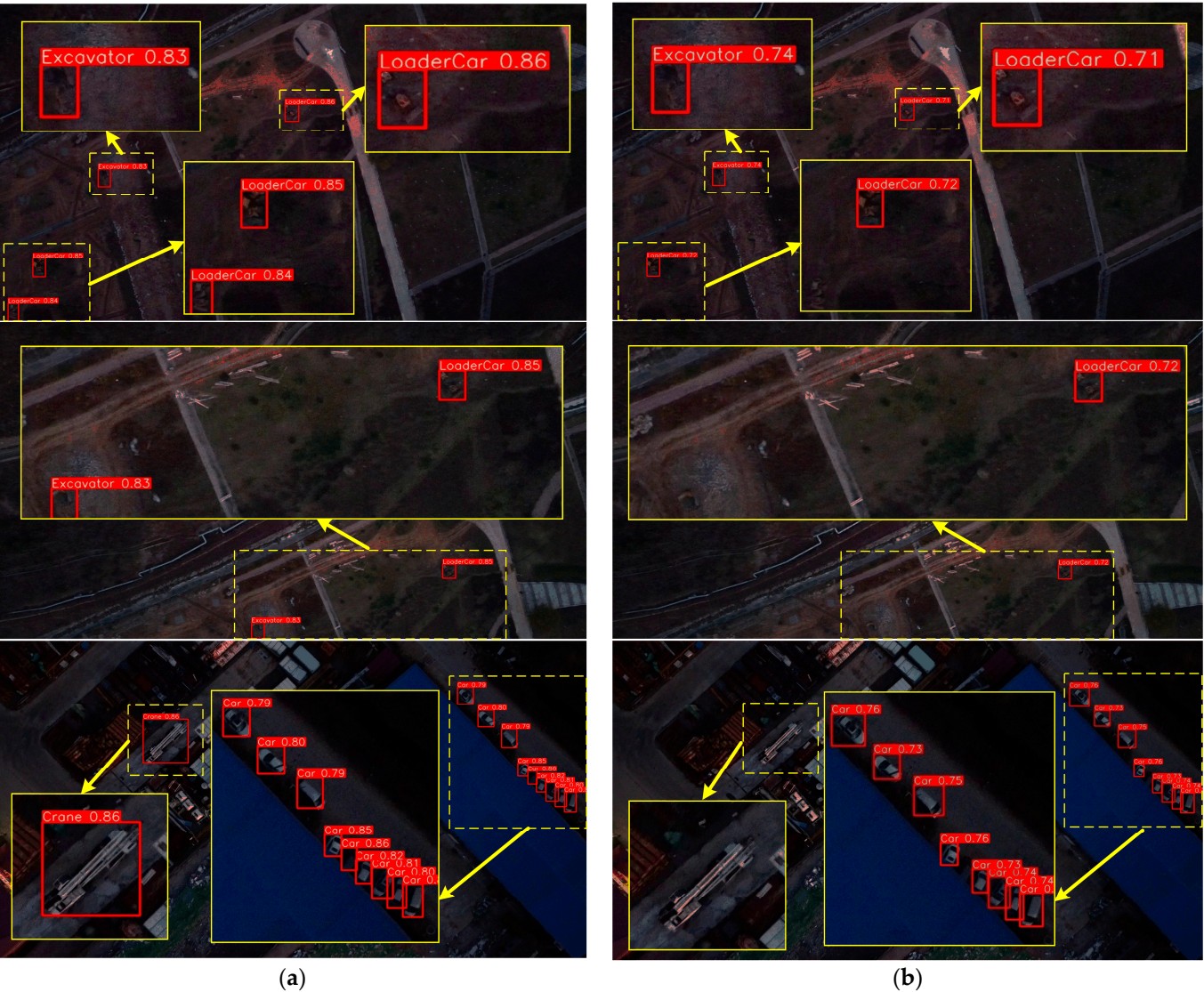

**Figure 15.** Recognition results in adverse light environment of SEVE Dataset. (**a**) Recognition results of the YOLO-GNS network; (**b**) Recognition results of the YOLO-V7 network.

3.3.2. Experiments on COCO Datasets

The evaluation metrics are mAP0.5, mAP0.75, and mAP0.5:0.95. mAP0.5 and mAP0.75 are the average accuracy of all target categories calculated at IOU thresholds of 0.5 and 0.75. mAP0.5:0.95 is the average accuracy of 0.5 to 0.95 at 0.05 intervals of 10. mAP0.5:0.95 is the average accuracy at 10 threshold values from 0.5 to 0.95 at 0.05 intervals.

As shown in Table 2, the experimental data show that the method in this paper also works well on the COCO dataset. The mAP0.5:0.95 is improved by 0.1% for YOLO-GNS compared to the original method with a similar speed. The mAP0.5 of YOLOV4 reaches 65.7% under this dataset; the mAP0.5 of YOLOV5-X is 68.8% under this dataset, but both networks are based on Darknet and its improvements with complex structures, and the detection speed is slightly lower than that of the present method. YOLO-GNS has 0.2% lower mAP0.75 than YOLOV7 on the COCO dataset but 0.1% higher mAP0.5; YOLO-GNS has improved detection speed and higher mAP than YOLOV4 and YOLOV5-X methods, indicating that the method in this paper is still effective on the public dataset COCO.

**Table 2.** Experimental results on coco dataset.

| Methods | Backbone | mAP0.5:0.95 | mAP0.5 | mAP0.75 |
|---|---|---|---|---|
| Faster-RCNN | ResNet50 | 36.2 | 59.2 | 39.1 |
| RetinaNet | ResNet50 | 36.9 | 56.3 | 39.3 |
| YOLOV4 | CSPDarknet-53 | 43.5 | 65.7 | 47.3 |
| YOLOV5-X | Modified CSP v5 | 50.4 | 68.8 | - |
| YOLOV7 | E-ELAN | 51.4 | 69.7 | 55.9 |
| YOLO-GNS | GhostELAN | 51.5 | 69.8 | 55.7 |

3.3.3. Ablation Experiment

Ablation experiments were conducted on the SEVE dataset to verify the effect of different network structures on the final detection results, and the experimental results are shown in Table 3.

**Table 3.** Results of ablation experiments.

| Methods | Backbone | GhostNet | SSH | mAP@0.5(%) |
|---|---|---|---|---|
| YOLOV7 | E-ELAN | × | × | 75.7 |
| YOLOV7 | E-ELAN | × | √ | 78.9 |
| YOLOV7 | E-ELAN | √ | × | 79.2 |
| YOLOV7 | E-ELAN | √ | √ | 80.1 |

"×" means no addition, "√" means addition.

With the addition of GhostNet in YOLOV7, the mAP value is improved by 3.5%. GhostNet forms the backbone network by forming GhostMP and GhostELAN modules, which has the advantages of maintaining the recognition performance of similarity and reducing the convolution operation at the same time and continuing to effectively increase the exploitation of feature maps, which is beneficial to the recognition of small targets. The addition of SSH structure in YOLOV7 improves the mAP value by 3.2%. SSH contextual network structure incorporates more concrete information and enhances the recognition of multiple details of special vehicles by increasing the perceptual field of the features, thus improving the detection performance. After adding both GhostNet and SSH structures in YOLOV7, the AP increases by 4.4%, further demonstrating that GhostNet and SSH can improve detection accuracy.

**4. Discussion**

The evaluation metrics examined in this study were AP and mAP. In the modified network, the values obtained from these criteria were as follows. The AP of cranes was 85.9%, the AP of loader cars was 86.9%, the AP of tank cars was 89.4%, the AP of mixer cars was 91.3%, the AP of forklifts was 90.1%, the AP for piling machines is 89.6%, the AP for road rollers is 69.5%, the AP for elevate cars is 67.3%, and the AP for excavators is 50.8%. Based on the basic results of the YOLOv7 network, it can be said that the proposed network has improved on average by 4.4% in accuracy and 1.6 in FPS, indicating that the improved network has improved speed to some extent with improved accuracy.

In recent years, the employment of artificial intelligence and deep learning methods has become one of the most popular and useful approaches in object recognition. Scholars have made many efforts to better detect vehicles in the context of UAV observations. Jianghuan Xie et al. proposed the residual feature enhanced pyramid network (RFEPNet), which uses pyramidal convolution and residual connectivity structure to enhance the semantic information of vehicle features [16]. One of the problems of these studies is the inability to detect small vehicles over long distances. Zhongyu Zhang et al. used a YOLOv3-based deep separable attention-guided network (DAGN), improved the loss function of YOLOv3, and combined feature tandem and attention blocks to enable the model to distinguish between important and unimportant vehicle features [19]. One of

the limitations of this study is the lack of types of vehicles and the lack of challenging images. Wang Zhang et al. helped the feature pyramid network (FPN) to handle the scale variation of vehicles by using the multi-scale feature adaptive fusion network (MSFAF-Net) and the region attention-based three-headed network (RATH-Net) [20]. However, the study did not address the crowded background images, hidden regions, and vehicle target-sensor distance, etc. Xin Luo et al. constructed a vehicle dataset for target recognition and used it for vehicle detection by an improved YOLO [21], but the dataset did not include special vehicles.

Previous research has focused on general vehicle detection, with a few studies examining the identification of different types of vehicles. In addition, the challenges of specialty vehicle identification, such as the small size of vehicles, crowded environments, hidden areas, and confusion with contexts such as construction sites, have not been comprehensively addressed in these studies. Thus, it can be argued that the unauthorized presence of specialty vehicles in challenging environments and the inaccurate identification of sensitive infrastructures remain some of the most important issues in ensuring public safety. The main goal of this study is to identify multiple types of specialty vehicles and distinguish them from ordinary vehicles at a distance, despite challenges such as the small size of specialty vehicles, crowded backgrounds, and the presence of occlusions.

In this study, the YOLOV7 network was modified to improve the challenges of specialty vehicle identification. A large number of visible images of different types of special vehicles and ordinary vehicles at close and long distances in different environments were collected and labeled to identify multiple types of special vehicles and distinguish them from ordinary vehicles. Considering the limited computational power of the airborne system, GhostNet is introduced to reduce the computational cost of the proposed algorithm. The proposed algorithm facilitates the deployment of airborne systems by using linear transformation to generate feature maps in GhostNet instead of the usual convolutional computation and requires less FLOP. On the other hand, the SSH structure is shown to have the ability to improve the detection accuracy of the algorithm. The context network is able to compute the contexts of pixels at different locations from multiple subspaces, which facilitates YOLO-GNS to extract important features from large-scale scenes. For example, in Figure 13, there are examples of special vehicles that the basic model cannot recognize in some cases. However, the modified model is able to recognize them; moreover, in other cases, they operate with lower accuracy than the modified network. This result indicates that the current network has improved in identifying special vehicles compared to the basic network. By applying these changes in the network structure and using a wide range of data sets, the proposed method is able to identify all specialty vehicle types in challenging environments. In Figures 14 and 15, examples of difficult images and poor lighting conditions are provided, all of which have higher recognition accuracy in the modified network than in the basic network.

## 5. Conclusions

As already pointed out, specialty vehicle recognition in various scenarios is a complex process; the usual approaches and even traditional deep learning network methods do not work well in some cases. When using UAVs to detect small or obscured specialty vehicles from large-scale scenes, both detection accuracy and computational consumption need to be considered. In this work, we propose a novel UAV-based algorithm for special vehicle target detection that enhances feature extraction while optimizing the feature fusion computation. A dedicated dataset of 17,992 UAV image datasets including multiple types of special vehicles is introduced, and extensive comparative experiments are conducted to illustrate the effectiveness of the proposed algorithm. The results show that the AP and FPS are improved by 4.4% and 1.6, respectively, compared to the primary YOLOv7. It can be demonstrated that the algorithm provides a single optimal solution for UAV-based target detection in the field of special vehicle identification. In the next work, the special vehicle detection method with visible and infrared fusion will be investigated.

**Author Contributions:** Conceptualization, Z.Q. and H.B.; Data curation, H.B.; Formal analysis, Z.Q. and H.B.; Funding acquisition, T.C.; Investigation, H.B.; Methodology, Z.Q.; Project administration, T.C.; Resources, T.C.; Software, Z.Q.; Supervision, H.B. and T.C.; Validation, Z.Q.; Visualization, Z.Q.; Writing—original draft, Z.Q.; Writing—review & editing, H.B. and T.C. All authors have read and agreed to the published version of the manuscript.

**Funding:** This work was supported in part by the National Key R&D Program of China (No.2022YFE0200300), the National Natural Science Foundation of China (No. 61972023), and the Beijing Natural Science Foundation (L223022).

**Data Availability Statement:** Not applicable.

**Conflicts of Interest:** The authors declare no conflict of interest.

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
