# Peer review of "Special Vehicle Detection from UAV Perspective via YOLO-GNS Based Deep Learning Network"

_drones, doi:10.3390/drones7020117_

Round 1
Reviewer 1 Report
The manuscript "Special Vehicle Detection from UAV Perspective via YOLO- 2 GNS-based Deep Learning Network" aims at presenting a novel algorithm for "special vehicle" detection from a UAV perspective. The authors introduce the single-stage headless (SSH) context structure to improve feature extraction and facilitate the detection of small or obscured objects while reducing computational costs through the use of GhostNet. The algorithm is tested on a dedicated dataset of 17,992 UAV images and compared to other derivatives, showing a 4.4% increase in average detection accuracy and a 1.6 increase in detection frame rate.
Overall, the manuscript presents a well-organized and well-researched study on special vehicle detection from a UAV perspective. The proposed algorithm is well explained, and the results of the experiments are clearly presented and discussed. The use of a dedicated dataset is also a positive aspect of the study. However, it would be beneficial for the authors to provide more detail on the dataset, such as the types of special vehicles included and the different scenarios in which the images were taken. Additionally, it would be useful for the authors to provide more information about the specific improvements in the detection of small or obscured objects, as well as a clearer explanation of the GhostNet technique used to reduce computational costs.
In summary, the manuscript is a valuable contribution to the UAV-based special vehicle detection field, and the proposed algorithm shows promising results. However, further details on the dataset and techniques used would help to strengthen the study.
Reviewer 2 Report
This paper presented a YOLO-GNS framework for special vehicle detection based on UAV images. Overall, the structure of this paper is well organized, and the presentation is relatively clear. The idea is interesting and has potential. However, there are still some crucial problems that need to be carefully addressed before a possible publication. More specifically,
1. UAV-based image target detection is actually in the field of remote sensing. Object detection in UAV RGB imagery is more challenging than general object detection due to variations in viewpoint and scale, lighting conditions, and high density of the objects. Thus, it would be better to combine the description of this reference (YOLOv5-Tassel: detecting tassels in RGB UAV imagery with improved YOLOv5 based on transfer learning) to highlight the significance of your work in the introduction.
2. You should provide the parameter size when you compare it with other models.
3. The text on many target boxes in the inference diagram is too small (such as Figure 11), please optimize it.
4. In YOLOv5, I remember that the weight of of mAP@0.5:0.95 is larger than AP@0.5. Would you mind explaining why you only select the AP@0.5 in Table 3?
Round 2
Reviewer 1 Report
N/A